# Immunoscore Predicted by Dynamic Contrast-Enhanced Computed Tomography Can Be a Non-Invasive Biomarker for Immunotherapy Susceptibility of Hepatocellular Carcinoma

**DOI:** 10.3390/cancers17060948

**Published:** 2025-03-11

**Authors:** Eisuke Ueshima, Keitaro Sofue, Shohei Komatsu, Nobuaki Ishihara, Masato Komatsu, Akihiro Umeno, Kentaro Nishiuchi, Ryohei Kozuki, Takeru Yamaguchi, Takanori Matsuura, Toshifumi Tada, Takamichi Murakami

**Affiliations:** 1Department of Radiology, Kobe University Graduate School of Medicine, Kobe 650-0017, Hyogo, Japan; ueshimae@med.kobe-u.ac.jp (E.U.); knishiuchi1002@gmail.com (K.N.); kozuki1023rp@gmail.com (R.K.); murataka@med.kobe-u.ac.jp (T.M.); 2Department of Surgery, Division of Hepato-Biliary-Pancreatic Surgery, Kobe University Graduate School of Medicine, Kobe 650-0017, Hyogo, Japan; komasho8@med.kobe-u.ac.jp (S.K.); nobu0527.kuro@gmail.com (N.I.); 3Department of Diagnostic Pathology, Kobe University Graduate School of Medicine, Kobe 650-0017, Hyogo, Japan; mkomatsu@hyogo-cc.jp; 4Division of Gastroenterology, Department of Internal Medicine, Kobe University Graduate School of Medicine, Kobe 650-0017, Hyogo, Japan; tmatsu@med.kobe-u.ac.jp (T.M.); tadat0627@gmail.com (T.T.)

**Keywords:** hepatocellular carcinoma, CECT, biomarker, tumor immune microenvironment, immunoscore, immunotherapy

## Abstract

Immunotherapy is the core treatment for hepatocellular carcinoma; however, the therapeutic efficacy is heterogeneous due to varying tumor immune microenvironments. This study aimed to noninvasively identify tumors that are more likely to benefit from immunotherapy by assessing the immune microenvironment of surgical resection sections and contrasting them with easily accessible CT findings. The results of this study showed that masses with peritumoral enhancement in the arterial phase have a higher immunoscore and are more likely to be susceptible to immunotherapy, which has a significant impact on clinical practice.

## 1. Introduction

Hepatocellular carcinoma (HCC) is often detected at an advanced stage, and immunotherapy with immune checkpoint inhibitors (ICIs) is the primary treatment option [1]. The mechanism of ICIs is to release the brakes on tumor immunity and help T cells attack the tumor; therefore, they are known to be highly dependent on the immune status within the tumor, particularly on the number of T cells [2].

The immunoscore proposed by Galon et al. predicts early-stage recurrence risk by quantifying the host CD3^+^ and CD8^+^ T cell response in the tumor interior (TI) and invasive margin (IM) [3] and is associated with prognosis and combined immunotherapy efficacy [4,5]. Therefore, the immunoscore may help determine the optimal treatment strategy for a growing number of systemic therapies. Although a tissue sample is required to calculate the immunoscore, liver tumor biopsy in patients with HCC is not recommended as a routine clinical procedure because of the risk of bleeding [6]. Overall, as a method for noninvasive pretreatment evaluation, the immunoscore is crucial for optimizing the outcome of patients treated with ICIs.

There have been several reports of MRI findings that predict immune status in HCC [7,8,9,10]. However, MRI is less accessible to patients, and there is a need for more accessible imaging techniques, such as computed tomography (CT). The association between immunoscore and preoperative contrast-enhanced CT (CECT) findings in patients with HCC has not yet been explored. Therefore, this study aimed to explore the ability of CECT features to predict HCC immunoscores.

## 2. Materials and Methods

### 2.1. Study Design

This study was conducted in accordance with the principles of the Declaration of Helsinki and its amendments and was approved by the Institutional Review Board Committee of Kobe University Hospital (No. B220241), and the requirement for informed consent was waived. This retrospective study consisted of two parts: cohort 1 for the evaluation of the immunoscore and CECT features, and cohort 2 for the validation of the CECT features in clinical patients (Figure 1).

In Cohort 1, 104 patients with HCC who underwent CECT within three months before surgical resection (January 2018 to December 2020) were enrolled. Eight patients with tumors that were not measurable using CECT were excluded, and 96 patients were included in this study. When the patients had multiple lesions, the largest lesion was enrolled in the analysis. We investigated the relationship among tumor characteristics, immunoscores, and CECT features.

In cohort 2, we evaluated the identified CECT features to assess the efficacy of combined immunotherapies of atezolizumab (anti-programmed death-ligand 1 (PD-L1) antibody) with bevacizumab (anti-vascular endothelial growth factor antibody) (n = 27) and durvalumab (anti-PD-L1 antibody) with tremelizumab (anti-cytotoxic T-lymphocyte-associated protein 4) (n = 14). This cohort included patients with pathologically or radiologically diagnosed HCC who underwent CECT within three months before combined immunotherapy (October 2020 and December 2023). Patients with evaluable follow-up CECT and no history of other systemic therapy administration or local treatment within 1 month prior to CECT and evaluable lesions (>15 mm) were included in this study. In patients who met the above criteria, up to three nodules from the largest tumor were enrolled. In total, 81 nodules from 41 patients were obtained, and time to nodular progression (TTnP) was evaluated using the Response Evaluation Criteria in Solid Tumors. TTnP was defined as an increase of more than 20% in each nodule and was compared between nodules with and without the identified CECT feature.

### 2.2. Histological Analysis

Surgical liver tissues were fixed in 10% formalin and embedded in paraffin, and formalin-fixed embedded (FFPE) samples were made. FFPE samples were sectioned at 5 μm thickness and mounted on microscope slides. Hematoxylin and eosin staining was performed in one section, and the contiguous sections were immunofluorescently stained as follows: sections were heated at 120 °C for 10 min to facilitate antigen retrieval. Once deparaffinization and rehydration were completed, the slides were incubated with a 1% hydrogen peroxide solution. After washing and blocking with skimmed milk, the slides were incubated with primary antibodies against CD3 (monoclonal mouse anti-human; M7254, Agilent, Santa Clara, CA, USA) and CD8^+^ antibodies (monoclonal mouse anti-human; Code: 413201, HISTOFINE, Tokyo, Japan) overnight at 4 °C. Antibodies diluted 1:200 were utilized as the primary antibodies. The slides were rinsed with the buffer and incubated with a biotinylated secondary antibody (Ultra-Sensitive ABC Peroxidase Rabbit IgG Staining Kit, Thermo Fisher Scientific, Waltham, MA, USA). Tissue staining was visualized using the DAB substrate chromogen solution (Thermo Fisher Scientific). The slides were counterstained with hematoxylin and dehydrated.

One pathologist (M.K.) reviewed all slides to assess CD3 and CD8-positive lymphocyte counts and to determine the frequency of CD3^+^ and CD8^+^ T cells in the tumor interior (TI) and invasive margin (IM) samples. The IM comprised approximately 50% of the tumoral area (tumor cells and stroma) and approximately 50% of the peritumoral area, whereas the TI included the nonnecrotic regions. Five representative regions of both the TI and IM were captured with 200× high-power magnification per slide. The captured images were analyzed using ImageJ software v0.4.0 (National Institutes of Health) to count the CD3^+^ and CD8^+^ T cells. The median immune cell density was used to stratify patients into groups based on the degree of tumor infiltration. The cutoff thresholds for CD3^+^ and CD8^+^ cell densities were 273 and 217.5 cells/mm^2^, respectively [11]. Based on the established threshold, each patient was assigned a binary score (0 = low, 1 = high) for each immune cell type (CD3^+^ and CD8^+^) in each tumor region (TI and IM). The immunoscore for each patient was obtained by summing the four binary scores on a scale ranging from 0 to 4. Five core groups were defined: patients with low densities of CD3^+^ and CD8^+^ T cells in both tumor regions were classified as having zero points; patients with one high density for one marker were classified as having one point; and patients with two, three, or four of these two markers were classified as having two, three, or four points (Figure 2).

### 2.3. Image Analysis

Multiphasic CECT images were acquired using multidetector row CT scanners (Aquilion 64 One; Canon Medical Systems, Otawara, Japan; SOMATOM Force, Siemens Healthcare, Erlangen, Germany). An iodinated contrast material was injected into the antecubital vein using a mechanical power injector at a dose of 600 mgI/kg for a fixed duration of 30 s. Multiphasic CT images comprised unenhanced, early arterial (20 s), late arterial (35–42 s), portal venous (70 s), and equilibrium (180 s) phase images. A bolus-tracking technique was used to acquire early arterial-phase images immediately after the trigger threshold was achieved. All multiphasic images were reconstructed using a slice thickness of 5 mm.

Two radiologists (K.S. and E.U., with 23 and 19 years of experience in abdominal radiology, respectively) who were blinded to patient information independently reviewed all CECT scans to assess (i) tumor size; (ii) gross morphology type (separated into three categories: SN, simple nodular; SNEG, single nodular type with extranodular growth; and CMN, confluent multinodular); (iii) features defined in the Liver Imaging Reporting and Data System (LI-RADS) v2018 [12] (non-rim arterial phase hyperenhancement [non-rim APHE], washout, enhancing capsule, rim APHE, delayed enhancement); and (iv) non-LI-RADS features such as peritumoral enhancement in the arterial phase, defined as detectable enhancement in the arterial phase adjacent to the tumor border and later becoming isoattenuating compared with the background liver parenchyma. Heterogenous enhancement is defined as irregular or variegated enhancement with multiple focal and subcapsular enhancing areas during the postcontrast phases. Intralesional arteries are defined as the persistence of discrete arteries within the lesion [13], and intratumoral necrosis is defined as non-enhancing areas (Figure 3). If a patient had multiple lesions, the largest lesion (main tumor) was assessed.

### 2.4. Statistical Analyses

The data are presented as medians ± standard deviations. Statistical analyses were performed using univariate and multivariate linear regression analyses to identify independent predictors of the immunoscore. The Kaplan–Meier method and log-rank test were utilized to assess the differences in TTnP. Differences with *p* values of <0.05 were considered statistically significant unless otherwise indicated. The GraphPad Prism software version 10.4.1 (GraphPad Prism, San Diego, CA, USA) was used for all analyses.

## 3. Results

### 3.1. Cohort 1

#### 3.1.1. Characteristics

Cohort 1 included 96 patients with HCC who underwent resection. The median patient age was 72 years (33–91). Among them, 20 (20.8%) were female and 76 (79.2%) were male. The patient characteristics, immunohistochemical findings, and CECT findings are summarized in Table 1. Regarding the immunoscores, 41 (42.7%), 10 (10.4%), 22 (22.9%), 16 (16.7%), and 7 (7.3%) patients were classified as having 0, 1, 2, 3, and 4 points, respectively. Among the 96 nodules, 53 (55.2%), 28 (29.2%), and 15 (15.6%) were SN, SNEG, and CMN based on gross morphology type, respectively. The median tumor size was 56.6 (15–160) mm. Of the 96 nodules, 15 (15.6%) and 40 (41.7%) showed rim APHE and peritumoral enhancement, respectively, in the arterial phase.

#### 3.1.2. Peritumoral Enhancement of CECT Findings Could Predict Immunoscore

There was no significant relationship between tumor size, gross morphology, and immunoscore. HCCs with rim APHE and peritumoral enhancement in the arterial phase were significantly associated with the immunoscore (*p* = 0.009 and <0.001, respectively) in the univariate regression analysis. The multivariate regression analysis revealed that HCCs with peritumoral enhancement in the arterial phase were independent predictors of the immunoscore (*p* = 0.004, Table 2).

### 3.2. Cohort 2

#### Susceptibility to Combined Immunotherapy of Nodules with Identified CECT Findings

The CECT scans of patients in cohort 2 who received combined immunotherapy (Table 3) were analyzed, and the presence of peritumoral enhancement in each nodule was determined. The median age of the patients was 70 years (48–83), and among the cohort, there were 4 (20.0%) females and 32 (80.0%) males. The median tumor size was 39.7 mm. Of the 81 nodules, 39 were SN type, 25 were SNEG, and 17 were CMN. Based on the CECT results, 27 nodules showed peritumoral enhancement in the arterial phase. Responses to combined immunotherapy included a complete response (CR) in 7 patients, partial response (PR) in 22 patients, stable disease (SD) in 38 patients, and progressive disease (PD) in 14 patients.

The median TTnP for the entire cohort was 294 days. Nodules with peritumoral enhancement (*p* < 0.001, median undefined vs. 180 days) demonstrated significantly prolonged TTnP compared to those without peritumoral enhancement (Figure 4).

## 4. Discussion

The findings of this study revealed that HCCs with peritumoral enhancement in the arterial phase (*p* = 0.004) were independent predictors of immunoscore. Furthermore, HCCs with this CECT feature were more likely to be susceptible to combined immunotherapy than those without this feature (*p* < 0.001).

Peritumoral enhancement in the arterial phase of gadolinium-ethoxybenzyl-diethylenetriamine pentaacetic acid-enhanced magnetic resonance imaging (EOB-MRI) has been reported to predict microvascular invasion and has been associated with early recurrence after surgery and locoregional treatment [14,15,16]. Furthermore, peritumoral enhancement on EOB-MRI has been shown to be associated with an immune-excluded phenotype and not immune-cold HCC [10]. Therefore, HCCs with peritumoral enhancement in the arterial phase could be included in the proliferative class and are consistent with an immune-rich microenvironment, as advocated by Llovet et al. [17]. Peritumoral enhancement reflects changes in hemodynamic perfusion around the nodules. Some studies have reported that the mechanism of peritumoral enhancement that fades to isoenhancement in the subsequent phases may be compensatory arterial hyperperfusion in the area of decreased portal venous flow caused by obstruction due to microscopic tumor thrombi around the tumor [18,19]. The direct relationship between the imaging features of peritumoral enhancement and the increased number of immune cells could not be investigated in the present study; however, further studies are warranted.

Immunoscore is a pathology-based assay for the quantification of CD3^+^ and CD8^+^ lymphocytes at the edge and core of a tumor [11,20]. By capturing the densities of both cell types in both regions, the immunoscore provided a scoring system ranging from low (immunoscore 0) to high (immunoscore 4). This immunoscore surpasses the classical TNM system for colorectal cancer stages I, II, and III in predicting DFS, disease-specific survival, OS, and overall survival [11]. This is in line with numerous reports confirming the positive prognostic value of CD8+ T cells not only in colorectal cancer but also in several other cancer types [3]. Immunoscores have also been reported as prognostic predictors of immunotherapy in colorectal cancer [21,22], renal cell carcinoma [23], and pancreatic cancer [24]. The current study demonstrates that tumors with peritumoral enhancement tend to have a higher immunoscore and a significantly better response to combined immunotherapy in nodal-based assessments. This is consistent with the contention that tumors with higher immunoscores have more tumor-infiltrating lymphocytes, which increase their response to combined immunotherapy [25,26,27]. Therefore, the results of this study allowed us to noninvasively predict susceptibility to immunotherapy based on easily accessible CT scan features, leading to personalized treatment.

This study has certain limitations that should be noted when interpreting our findings. The immunoscore was determined based on the number of CD3^+^ and CD8^+^ lymphocytes. Recently, a new immunoscore that includes the distance to PD-L1 was proposed [28,29], which could be used for future evaluation. Second, the validation cohort was treated with combined immunotherapy; however, new combinations of combined immunotherapies have been introduced [30] but have not yet been evaluated. Similar studies should be conducted on patients receiving new combination immunotherapy regimens. Third, our study did not consider differences in patient backgrounds, such as etiology or sex. However, we believe that this should be considered in future studies with larger numbers of patients and more diverse study cohorts. Finally, CT evaluations were categorical, which may lead to inaccuracies in the lesions on the binary strike-line. As a countermeasure, the parameter should be graded rather than given a binary assessment. Alternatively, it may be necessary to develop comprehensive criteria that include serological assessments.

## 5. Conclusions

HCCs with peritumoral enhancement in the arterial phase of contrast-enhanced computed tomography may provide noninvasive biomarkers for predicting the immunoscore and are more susceptible to combined immunotherapy than those without these features.

## Figures and Tables

**Figure 1 cancers-17-00948-f001:**
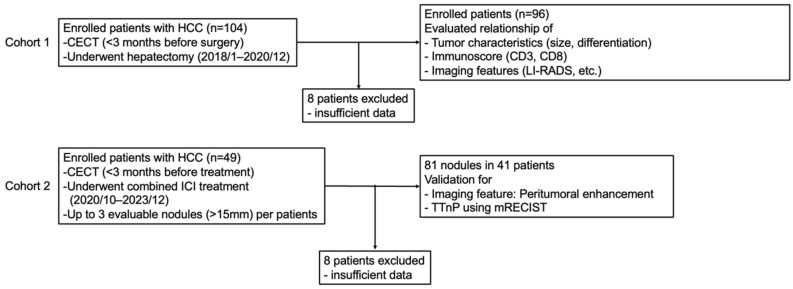
Flow diagram of this study. HCC, hepatocellular carcinoma; CECT, contrast-enhanced computed tomography; LI-RADS, Liver Imaging Reporting and Data System; TTnP, time to nodular progression; mRECIST, modified Response Evaluation Criteria in Solid Tumors.

**Figure 2 cancers-17-00948-f002:**
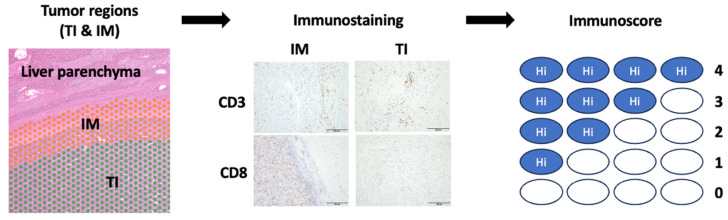
Immunoscore calculation. Tumor regions are manually divided into tumor interior (TI) and invasive margin. Immunohistochemical staining was performed with CD3^+^ and CD8^+^ antibodies. The CD3^+^ and CD8^+^ T cells in the tumor interior (TI) and invasive margin (IM) groups were counted, respectively. Based on the established threshold, the immunoscore was calculated for each immune cell type (CD3^+^ and CD8^+^) in each tumor region (TI and IM), from 0 to 4. Scale bar: 100 μm.

**Figure 3 cancers-17-00948-f003:**
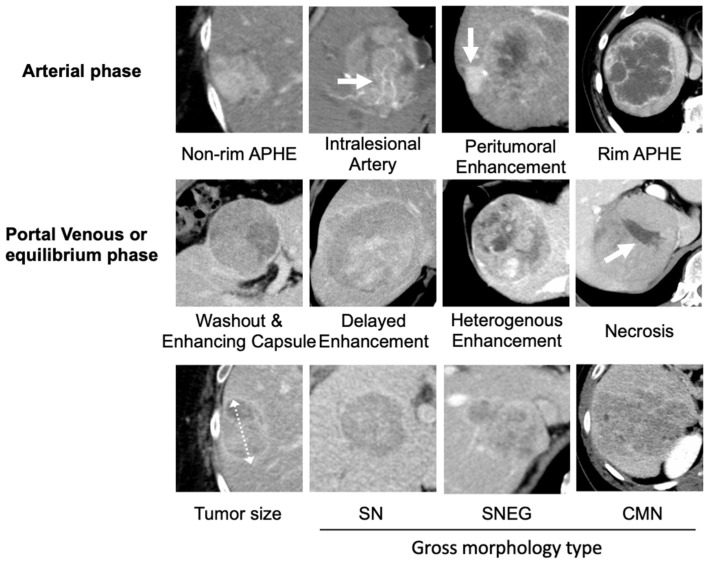
CECT findings evaluated in this study. APHE, arterial phase hyperenhancement; SN, simple nodule; SNEG, single nodular type with extranodular growth; CMN, confluent multinodular.

**Figure 4 cancers-17-00948-f004:**
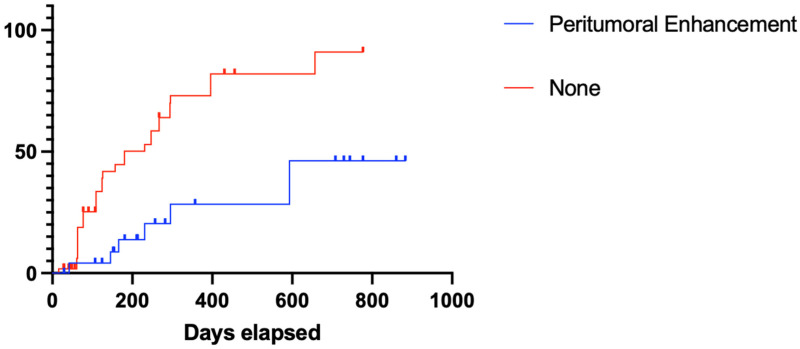
Time to nodular progression of HCCs with peritumoral enhancement treated with ICIs. TTnP: time to nodular progression, HCC, hepatocellular carcinoma; ICIs, immune checkpoint inhibitors.

**Table 1 cancers-17-00948-t001:** Characteristics in cohort 1.

Patient Characteristics	n = 96
Age [range, SD] (years)	72 [33–91, 9.9]
Sex (female/male)	20/76
Etiology (HBV/HCV/NBNC)	21/27/48
Alcohol usage	36
BMI [range, SD] (kg/m^2^)	23.2 [16.1–39.0, 4.1]
Diabetes mellitus	31
Liver cirrhosis	15
Child–Pugh (A/B/C)	82/12/2
BCLC stage (A/B/C)	49/30/17
Number of tumors [range, SD],	1 [1–10]
AFP [range, SD], (ng/mL)	9 [1.5–533,413, 65,978]
DCP [range, SD], (AU/L)	219 [15–473,139, 58,368]
Differentiation (wel/mod/por)	9/69/18
**IHC Findings**	**n = 96**
TI CD3	58 [3–921]
TI CD8	14 [0–585]
IM CD3	96 [8–950]
IM CD8	21 [0–671]
Immunoscore (0/1/2/3/4-points)	41/10/22/16/7
**CT Findings**	**n = 96**
Tumor size [range, SD] (mm)	57 [15–160, 38]
Gross morphology (SN/SNEG/CM)	53/28/15
Non-rim APHE	80
Washout	93
Enhancing capsule	78
Rim APHE	15
Peritumoral enhancement	40
Delayed enhancement	18
Heterogenous enhancement	69
Intralesional artery	48
Necrosis	34

BCLC, Barcelona Clinic Liver Cancer; AFP, alpha-fetoprotein; BMI, body mass index; DCP, des-gamma-carboxyprothrombin; HBV, hepatitis B virus; HCV, hepatitis C virus; NBNC, non-B non-C; IT, intratumoral; IM, invasive margin; APHE, arterial phase hyperenhancement.

**Table 2 cancers-17-00948-t002:** Predictors of immunoscore by linear regression analysis.

Variables	Estimate	Standard Error	95% CI	|t|	*p*-Value	VIF
Univariate						
Tumor size (mm)	0.034	0.037	0.107 to 0.039	0.932	0.354	-
Gross morphology (SN vs. SNEG/CM)	−0.547	0.318	−1.177 to 0.084	1.720	0.089	-
Nonrim APHE	−0.642	0.362	1.360 to 0.077	1.774	0.079	-
Washout	0.705	0.503	−0.294 to 1.703	1.401	0.165	-
Enhancing capsule	0.590	0.355	−0.114 to 1.294	1.663	0.100	-
Rim APHE	−1.002	0.373	−1.742 to −0.263	2.690	0.009 **	-
Peritumoral enhancement	−1.076	0.274	−1.621 to −0.532	3.928	<0.001 **	-
Delayed enhancement	0.528	0.383	−0.232 to 1.289	1.380	0.171	-
Heterogenous enhancement	0.047	0.294	−0.536 to 0.630	0.162	0.872	-
Intralesional artery	0.128	0.281	−0.430 to 0.686	0.456	0.650	-
Necrosis	−0.250	0.297	0.839 to 0.339	0.842	0.402	-
**Multivariate**						
Intercept	2.323	0.332	1.664 to 2.981	7.004	<0.001	
Rim APHE	−0.432	0.407	−1.240 to 0.375	1.063	0.290	1.289
Peritumoral enhancement	−0.920	0.311	−1.537 to −0.302	2.958	0.004 **	1.289

** Significant *p* values. APHE, arterial phase hyperenhancement; CI, confidence interval; VIF, variance inflation factor.

**Table 3 cancers-17-00948-t003:** Characteristics in cohort 2.

Patient Characteristics	n = 40 (Patients)
Age [range, SD], (years)	70 [48–83, 8]
Sex (female/male)	8/32
Etiology (HBV/HCV/NBNC)	7/11/22
Child–Pugh (A/B/C)	34/5/1
BCLC stage (A/B/C)	0/21/19
Number of tumors enrolled [range, SD],	2 [1–3]
ICI treatment (Ate + Bev/Dur + Tre)	26/14
**CT Findings**	**n = 81 (Nodules)**
Tumor size [range, SD], (mm)	39.7 [11–157, 35]
Gross morphology (SN/SNEG/CM)	39/25/17
Peritumoral enhancement	27
**Treatment Effect**	**n = 81 (Nodules)**
Best response CR/PR/SD/PD	7/22/38/14
TTnP median	294 days
Uncensored/censored	41/40

HBV, hepatitis B virus; HCV, hepatitis C virus; NBNC, non-B non-C; BCLC, Barcelona Clinic Liver Cancer; ICI, immune checkpoint inhibitor; TTnP, time to nodular progression; CR, complete response; PR, partial response; SD, stable disease; PD, progressive disease.

## Data Availability

The original contributions of this study are included in this article. Further inquiries can be directed to the corresponding authors.

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
