# Peer review of "Immunoscore Predicted by Dynamic Contrast-Enhanced Computed Tomography Can Be a Non-Invasive Biomarker for Immunotherapy Susceptibility of Hepatocellular Carcinoma"

_cancers, 2025, doi:10.3390/cancers17060948_

Round 1

Reviewer 1 Report

Comments and Suggestions for Authors

I would like to congratulate the researchers team for presenting a very interesting study demonstrating that HCC tumours with peritumoral enhancement in the arterial phase on CECT have a higher immunoscore and are more likely to be susceptible to immunotherapy, which has a significant impact on clinical practice. There is a great need for more non-invasive and imaging based prognostic tools to stratify patients with advanced cancers for personalized treatments, including cytoreductive surgery, immunotherapy and other targeted precision therapies. This study demonstrates a very interesting and applicable in current clinical practice prognostification model, which needs to be tested in more clinical trials. However, the approach is novel and could potentially be investigated in other malignancies. Certainly the paper is of interest to many readers and it meets the requirements for the publication in this type of journal. Introduction is concise and clear. Methodology is clear and appropriate. Results are presented in a clear and easily understandable manner. The discussion is comprehensive and based on recent literature. The conclusions are based in the results of the study.

Author Response

Response: Thank you for your encouraging comments.

Reviewer 2 Report

Comments and Suggestions for Authors

This report generates a useful method to identify HCC.

Major concerns and comments:

  1. The report needs to add more patients' information including history of alcohol usage and smoking, BMI/BRI, diabetes or not, etc.
  2. Table 1: please add standard deviation or error on Age   group, AFP group, Tumor size group.
  3. Table 1: what does "Numbers of tumors " mean? What is "1[1-10]"?
  4. Table 3: what does "Numbers of tumors enrolled " mean? What is "2[1-3]"?
  5. The current study does not address the difference between gender. Any females vs males difference?
  6. The reference list is appropriate.
  7. Please list all abbreviations.
  8. Any other countries or reports have similar reports?

Author Response

This report generates a useful method to identify HCC.

Major concerns and comments:

1. The report needs to add more patients' information including history of alcohol usage and smoking, BMI/BRI, diabetes or not, etc.

Response: Thank you for your comment. We added those information in the manuscript mainly in the Table 1.

2. Table 1: please add standard deviation or error on Age group, AFP group, Tumor size group.

Response: Thank you for pointing them out. We added SD in the Table 1.

3. Table 1: what does "Numbers of tumors " mean? What is "1[1-10]"?

4. Table 3: what does "Numbers of tumors enrolled " mean? What is "2[1-3]"?

Response: We apologize for the lack of information. We have added an annotation in the parentheses both in Table 1 and 3, respectively.

We have also added the following sentence in line 94-95; “In patients who met the above criteria, up to three nodules were enrolled in order from the largest tumor.”

5. The current study does not address the difference between gender. Any females vs males difference?

Response: Thank you for bringing this issue to our attention. We think it would be interesting to examine differences in the tumor immune environment by gender. Since this study focused on differences in imaging findings, we did not examine differences in patient background. This should be considered in future studies. We have added the above content to the limitation section as follows;

Line 281-283, “Thirdly, our study did not take into account differences of patient background, such as etiology or gender. We believe this should be considered in future studies with larger numbers of patients”

6. The reference list is appropriate.

Response: Thank you for your comment.

7. Please list all abbreviations.

Response: Thank you for your comment. We have listed necessary abbreviations in the last of manuscript, line 293.

8. Any other countries or reports have similar reports?

Response: Thank you for your comment. As mentioned in the introduction, there are several studies on the prediction of the tumour immune environment using MRI imaging findings (ref 7-10). However, there are no published studies focusing on the relationship between CT imaging and Immunoscore that we could find.

Round 2

Reviewer 2 Report

Comments and Suggestions for Authors

No more comments